# Infrared Thermography for Measuring Elevated Body Temperature: Clinical Accuracy, Calibration, and Evaluation

**DOI:** 10.3390/s22010215

**Published:** 2021-12-29

**Authors:** Quanzeng Wang, Yangling Zhou, Pejman Ghassemi, David McBride, Jon P. Casamento, T. Joshua Pfefer

**Affiliations:** 1Center for Devices and Radiological Health, Food and Drug Administration, Silver Spring, MD 20993, USA; zenobiachow@gmail.com (Y.Z.); Pejman.Ghassemi@fda.hhs.gov (P.G.); jon.casamento@fda.hhs.gov (J.P.C.); Joshua.Pfefer@fda.hhs.gov (T.J.P.); 2Department of Mechanical Engineering, University of Maryland, Baltimore County, Baltimore, MD 21250, USA; 3University Health Center, University of Maryland, College Park, MD 20742, USA; drmcb2595@gmail.com

**Keywords:** infrared thermography, elevated body temperature, fever screening, clinical accuracy

## Abstract

Infrared thermographs (IRTs) implemented according to standardized best practices have shown strong potential for detecting elevated body temperatures (EBT), which may be useful in clinical settings and during infectious disease epidemics. However, optimal IRT calibration methods have not been established and the clinical performance of these devices relative to the more common non-contact infrared thermometers (NCITs) remains unclear. In addition to confirming the findings of our preliminary analysis of clinical study results, the primary intent of this study was to compare methods for IRT calibration and identify best practices for assessing the performance of IRTs intended to detect EBT. A key secondary aim was to compare IRT clinical accuracy to that of NCITs. We performed a clinical thermographic imaging study of more than 1000 subjects, acquiring temperature data from several facial locations that, along with reference oral temperatures, were used to calibrate two IRT systems based on seven different regression methods. Oral temperatures imputed from facial data were used to evaluate IRT clinical accuracy based on metrics such as clinical bias ($\Delta_{cb}$), repeatability, root-mean-square difference, and sensitivity/specificity. We proposed several calibration approaches designed to account for the non-uniform data density across the temperature range and a constant offset approach tended to show better ability to detect EBT. As in our prior study, inner canthi or full-face maximum temperatures provided the highest clinical accuracy. With an optimal calibration approach, these methods achieved a $\Delta_{cb}$ between ±0.03 °C with standard deviation ($\sigma_{\Delta cb}$) less than 0.3 °C, and sensitivity/specificity between 84% and 94%. Results of forehead-center measurements with NCITs or IRTs indicated reduced performance. An analysis of the complete clinical data set confirms the essential findings of our preliminary evaluation, with minor differences. Our findings provide novel insights into methods and metrics for the clinical accuracy assessment of IRTs. Furthermore, our results indicate that calibration approaches providing the highest clinical accuracy in the 37–38.5 °C range may be most effective for measuring EBT. While device performance depends on many factors, IRTs can provide superior performance to NCITs.

## 1. Introduction

Fever is a key symptom of many infectious diseases that have produced epidemics, including Severe Acute Respiratory Syndrome (SARS) in 2003, Influenza A (H1N1) in 2009, Ebola Virus Disease (EVD) in 2014, and Coronavirus (COVID-19) in 2019–present [1,2,3,4,5,6]. While fever screening alone is not an effective method to stop an epidemic, it is likely that for many infectious diseases it can be part of a larger approach to risk management. In several recent epidemics, fever screening has been used in high-traffic areas and at the entrances of high-risk sites, such as public transportation hubs, hospitals, and assisted living facilities, yet there is little evidence that this approach has made a significant impact [7]. This may be due in part to the implementation of ineffective instrumentation and calibration algorithms, as well as a lack of viable, consistently applied standard procedures for deployment and screening.

Body temperature can be measured at different body sites. These measurements can be used to impute temperatures at other body sites that are more meaningful, but less convenient to access. The site where the temperature is acquired is called the measurement site, whereas the site to which the device output temperature refers is called the reference site. For example, a non-contact infrared thermometer (NCIT) might measure skin temperature on the forehead and convert this value to an imputed oral temperature for display. In this case, the forehead-center is the measurement site and the oral cavity (e.g., sublingual) is the reference site. The process of imputing reference site temperature from measurement site temperature is called site conversion. The measurement and reference sites can be the same (same-site measurement) or different (cross-site measurement).

Through autonomic physiological mechanisms, humans can maintain internal temperature (also known as core body temperature) within very narrow limits despite wide fluctuations in ambient air temperature, so as to ensure proper physiological function [8]. Human thermoregulation processes include chemical reactions, perfusion inside the body, and heat transfer with the environment through radiation, conduction, convection, and evaporation. Temperatures at different peripheral body sites can be quite different and have more fluctuation due to factors such as ambient temperature [9,10], exercise [11], metabolic rate [12], circadian rhythm [13,14], age [15], and menstrual cycle [16]. Therefore, it is difficult to accurately define the relation between temperatures at two different body sites with a mathematical model due to the complexity of human thermoregulation mechanisms. Thus, the accuracy of output temperature from a cross-site measurement is often lower than that from a same-site measurement, since imputing the reference site temperature from the measurement site temperature will increase cumulative error.

NCITs [17,18] and infrared thermographs (IRTs, also known as thermal cameras) [19] represent the primary device types currently used in practice for fever screening during epidemics. IRTs and NCITs use similar principles for temperature measurement. Although NCITs are highly portable, inexpensive, and have been widely used for fever screening during epidemics [20], their accuracy has been called into question, particularly relative to IRTs [21,22]. This may be due to a range of factors including the common use of forehead measurement locations, which tend to be more susceptible to fluctuations due to environmental factors like ambient temperature and airflow [23]. The effectiveness of prior IRT-based approaches to reduce the spread of disease has also been mixed. While some human subject studies demonstrated that IRTs can estimate body temperature with moderately high accuracy [21,24,25,26], others indicated that IRTs are not effective for fever screening [27,28,29]. In many situations, it may not be practical to implement all of the required controls necessary to ensure a high degree of thermal screening performance. Low IRT effectiveness may also be attributable in part to the use of IRTs with insufficient performance specifications, improper deployment practices [30,31], and/or a lack of febrile subjects in clinical studies.

Laboratory accuracy [32] is a key performance characteristic of IRTs. International standard IEC 80601-2-59:2017 provides recommendations for laboratory accuracy evaluation of fever-screening IRTs [30]. However, clinical accuracy determined from a clinical study is much more relevant since it incorporates real-world variability due to the device, subjects and environment, as well as the temperature conversion step between measurement and reference sites. Currently, there are no consensus methods to evaluate the clinical accuracy of IRTs. A technical report, ISO/TR 13154:2017 [31], describes best practices for IRT deployment, implementation and operation, yet evaluation of IRT clinical accuracy is not covered. Two international standards which address methods to evaluate the clinical accuracy of thermometers, namely ASTM E1965-98:2016 [33] and ISO 80601-2-56:2017 [34], provide relevant insights, yet they have not been adapted for use in IRT performance testing.

During clinical studies, temperatures should be measured both with the IRT on the face and a clinical thermometer with established clinical accuracy at the reference site. While the literature indicates that a number of internal tissue sites, including the pulmonary artery [35], esophagus, urinary bladder, and rectum [36], are suitable for estimating core temperature, they are impractical for large-scale clinical fever screening studies. Tympanic membrane and oral cavity thermometry are often used, however, the former approach has shown poor performance in some studies because of dirt/cerumen, inaccurate placement and lack of skill of the measurer [36,37,38]. Oral thermometry provides a well-correlated surrogate location for core temperature and is not very susceptible to confounding factors [36,39,40].

In our recent prior article [41], we provided an initial analysis of our clinical study data, focusing on the 596 subjects measured within the room temperature range of 20–24 °C. In the current work, we have analyzed the entire dataset of more than 1000 subjects measured within the room temperature range of 20–29 °C. Our primary intent of this study was to compare methods for IRT calibration based on clinical data and identify best practices for assessing the clinical performance of IRTs intended to detect elevated body temperatures (EBT). A key secondary aim was to compare IRT clinical accuracy to that of NCITs. Specifically, we (a) acquired IRT and reference temperature data in febrile and non-febrile subjects using methods that closely adhered to international standards, (b) analyzed the relationship between reference temperature and facial temperatures at different locations, (c) evaluated the impact of different training/calibration techniques on clinical accuracy, (d) compared different metrics as clinical accuracy indicators, and (e) compared results to similar data from NCITs.

## 2. Methods

Over the course of 18 months, from November 2016 to May 2018, we conducted a clinical study at the Health Center of the University of Maryland (UMD) at College Park according to the guidelines of the Declaration of Helsinki. The study was approved by both FDA and UMD Institutional Review Boards under FDA IRB study #16-011R and written informed consent was obtained from all subjects.

### 2.1. Experimental Setup and Temperature Measurement Procedure

The primary devices used included an oral thermometer (SureTemp Plus 690, Welch Allyn, San Diego, CA, USA) with established clinical accuracy, a webcam (C920, Logitech, Lausanne, Switzerland), two IRTs (IRT-1: 320 × 240 pixels, A325sc, FLIR Systems Inc., Nashua, NH, USA; IRT-2: 640 × 512 pixels, 8640 P-series, Infrared Cameras Inc., Beaumont, TX, USA), a blackbody (SR-33, CI Systems Inc., Carrollton, TX, USA) as the external temperature reference source (ETRS) for temperature drift compensation, and six models of NCITs. The laboratory accuracy of both IRT systems satisfied the IEC 80601-2-59:2017 standard requirements [30] in terms of stability, drift, minimum resolvable temperature difference, and radiometric temperature laboratory accuracy, as shown in our previous study [32]. An IRT system (also known as a screening thermograph) is composed of an IRT and an ETRS. [30,32]. For brevity, we call an IRT system an IRT in this paper.

The study lasted for 18 months covering all four seasons, which can explain why we had a wide ambient temperature range of 20–29 °C due to inefficient air conditioning in summer. To minimize the influence of outside temperature, each subject was preconditioned by waiting for at least 15 min in the draft free study area inside the building before starting the measurements. For each subject, four rounds of measurements were performed within ~15 min. During each round, temperatures were measured with two different IRTs, six models of NCITs and a contact oral thermometer.

The IRTs used skin emissivity and ambient temperature as input parameters to calculate skin temperature automatically. Publications have suggested that the emissivity values of the anterior surface of the eyeball and skin are 0.975 [42] and 0.98 [43,44], respectively. Therefore, skin emissivity of 0.98 was used as an IRT input parameter, which is also recommended by the IEC 80601-2-59:2017 standard [30]. The ambient temperature was also measured with a weather tracker prior to each measurement as an IRT input parameter. We did not perform any other laboratory calibration/correction except for the temperature compensation with an ETRS (see Section 2.3.1 in our previous publication [41] for details; the ETRS emissivity value of 0.98 was used in our algorithm as suggested by the manufacturer).

Temperature measured with the contact oral thermometer was used as the reference ($T_{ref}$). NCIT measurements performed in this study are addressed in greater depth elsewhere [45]. Additional information about the study methods (e.g., device setup, environmental control, measurement procedure) can be found in our published paper [41]. Ideally, the ambient temperature should be 20–24 °C and relative humidity 10–50%, based on the ISO/TR 13154 document [31]. In our study, however, ambient temperature was between 20 and 29 °C, and relative humidity was between 10% and 62% (Figure 1). While beyond the recommended ranges, these conditions more realistically emulate real-world fever screening settings.

### 2.2. Subject Demographics

Data were acquired and analyzed from a total of 1020 subjects for IRT-1 and 1010 subjects for IRT-2. Demographic information for study subjects is summarized in Table 1. Overall, about 11% of these subjects exhibited reference temperature above 37.5 °C.

### 2.3. Facial Region Delineation and Temperature Measurement

We identified facial key-points in IRT images by matching landmarks on visible light images to thermal images with an image registration approach [46] as well as manual labeling. Based on the identified facial key-points, different regions/points on thermal images were defined and the temperatures at these regions were obtained from thermal images (Figure 2). Since IRTs exhibit varying degrees of instability and drift [32], all IRT-measured temperatures were compensated with a blackbody (ETRS) in the system. Details about the definitions of these temperatures and temperature compensation with an ETRS can be found in Section 2.2 and Section 2.3.1, respectively, in our previous publication [41].

For brevity, we restricted our analysis to four main facial temperatures ($T_{skin}$): $T_{FC}$, $T_{FCmax}$, $T_{CEmax},$ and $T_{max}$. Inner canthi are considered to be optimal locations for non-contact temperature measurement [30]. Perfused by the internal carotid artery, they are typically the warmest regions on the face and have high stability and strong correlation with internal body temperatures [19,47,48]. However, there is no consensus about how canthi temperature should be read (e.g., how to identify location, size of region to use, number of pixels, averaging vs. maximum value, etc.). Among all the temperatures obtained from the inner canthi region, our initial study demonstrated that $T_{CEmax}$, the maximum temperature of the extended canthus region (see Figure 2), has the best correlation with the reference oral temperature $T_{ref}$ and the highest sensitivity (Se) and specificity (Sp) values for fever screening [41]. Therefore, we chose $T_{CEmax}$ for further study in this paper. Our previous work also demonstrated that the whole face maximum temperature ($T_{max}$) is easy to localize/calculate and has comparable performance to $T_{CEmax}$, especially considering that for 59.5% of subjects, $T_{max}$ and $T_{CEmax}$ have the same location. Please see reference [41] for the distribution of thermal maxima in full-face images. Since many NCITs measure temperature from the forehead-center location with a small sensor, $T_{FC}$ measured with an IRT was used as a surrogate for NCITs. Other NCITs use a sensor array to detect temperature in a larger forehead region; $T_{FCmax}$ was used as a surrogate for such devices since a similar region is detected.

### 2.4. Clinical Data

Data from 1115 subjects were originally collected. Of these, 6 subjects had incomplete records. The data for 56 subjects were also removed because the difference between the two oral temperature readings was greater than 0.5 °C, or only one oral temperature reading was recorded. The large difference might come from an operation error (e.g., oral thermometer moved) or the subjects have recently smoked or ingested cold or hot food or drink [49]. Of the remaining subjects, we further excluded 33 subjects for IRT-1 and 43 subjects for IRT-2 whose images had degraded quality due to motion artifacts. Finally, we had data from 1020 subjects measured with IRT-1 and 1010 subjects measured with IRT-2.

The data for each IRT were separated into two groups—Group 1 with ambient temperature ranged from 20 to 24 °C and Group 2 from 24 to 29 °C (Table 2). The temperature ranges are different because the clinical study lasted a long time at two different locations (a small room and hallway), resulting in large ambient temperature variation. Group 1 data were first analyzed in our prior work [41], since ISO/TR 13154:2017 [31] recommends ambient temperature range of 20–24 °C. We analyzed Group 2 data with the same methodology as Group 1 data analysis in terms of the correlation coefficients and the area under the curve (AUC) values for different receiver operator characteristic (ROC, described further in Section 2.6.2) curves. The results show that both groups have similar performance in terms of correlation coefficients (Table 3) and AUC values (Table 4). In this study, we evaluate IRT clinical accuracy with more metrics than our previous analysis, which needs larger amount of data for calibration and testing. Therefore, both Group 1 and Group 2 data were used in the current paper.

### 2.5. Regression Methods for Imputing Oral Temperature

Many IRTs convert measured skin temperature ($T_{skin}$) to an imputed corresponding temperature at a reference body site [34], often sublingual oral temperature ($T_{oral}$), which is called cross-site measurement in this paper. In this study, we evaluated the clinical accuracy of two IRTs based on a cross-site measurement approach. Data acquired for each subject include thermal images, NCIT readings (analyzed in [45]) and reference sublingual temperature ($T_{ref}$). Thermal images were used to extract $T_{skin}$ at different regions of interest ($T_{FC}$, $T_{FCmax}$, $T_{CEmax}$ and $T_{max}$). The conversion from $T_{skin}$ to $T_{oral}$ required the use of a calibration curve, so subjects for each IRT were randomly separated into training and testing sets. The training set (60% of the subjects, 612 and 606 for IRT-1 and IRT-2 respectively) was used to establish the relationship between different $T_{skin}$ and $T_{ref}$. The testing set (remaining 40% of subjects, 408 and 404 for IRT-1 and IRT-2 respectively) was converted to $T_{oral}$ values based on the calibration curve, then compared with $T_{ref}$ to evaluate clinical accuracy.

The relationship between $T_{skin}$ and $T_{ref}$ can be determined with different regression methods. In our previous study [41], we observed that $T_{skin}$ and $T_{ref}$ appear to be related by a constant offset or a linear relation. Therefore, constant offset and ordinary linear regression methods are applied here. Quadratic or higher order polynomial regressions are also considered. Since $T_{ref}$ values likely contain significant error, Deming regression may also be appropriate [50].

Since the distribution of $T_{ref}$ values is not uniform across the temperature range (See the Kernel density curves in Section 3.1), with significantly less data at low and high temperatures, three regression approaches were considered. Weighted linear regression is a technique that adjusts the influence of individual data points based on a predefined criterion [50]. Common weighting methods are often based on variance or coefficient of variation (CV). For example, a constant CV least-squares regression gives each point a weight inversely proportional to the square of the values on the *x*-axis [50]. We implemented a weighted regression method with the weight being inversely related to the kernel density of the independent variable, i.e., greater weight was applied to a temperature range with fewer data points. A second approach implemented, called a binning method here, involved dividing the training data into small intervals (“bins”) and the data in each interval are averaged as one value for regression. A third approach used to mitigate the uneven data distribution was segmented linear regression, also known as piecewise regression. In this method, training data were separated into several segments and linear regression is applied to each. The equations for each segment were forced to agree at the edges to ensure continuity.

### 2.6. Clinical Accuracy Assessment

The clinical accuracy of IRTs can be evaluated in two ways. One way is to see whether IRTs can accurately measure body temperature in a specific temperature range, called temperature measurement accuracy in this paper. The other way is to see whether IRTs can screen out subjects with EBT from those without EBT, called diagnostic performance in this paper.

#### 2.6.1. Metrics for Temperature Measurement Accuracy

We evaluated the temperature measurement accuracy of IRTs using several different approaches. Since there is no standard that covers clinical study data analysis for IRTs, standards for thermometers were used to inform our methodology. The standards ISO 86601-2-56:2017 [34] and ASTM E1965-98:2016 [33] implement three key metrics: clinical bias ($\Delta_{cb}$), standard deviation (SD) of $\Delta_{cb}$ ($\sigma_{\Delta cb}$), and clinical repeatability ($\sigma_{r}$). $\Delta_{cb}$ is the mean difference between $T_{oral}$ and $T_{ref}$ values for all subjects in the testing set. It shows systematic error of the devices under test. Measurement precision was evaluated using $\sigma_{\Delta cb}$, which is based on the SD of differences between $T_{oral}$ and $T_{ref}$. A value equal to 2 × $\sigma_{\Delta cb}$ is often called the limit of agreement ($L_{A}$), as it shows the magnitude of potential disagreement between outputs of two devices when used on the same human subject. Difference plots are used to illustrate $\Delta_{cb}$ and $\sigma_{\Delta cb}$.

Root-mean-square (RMS) difference ($A_{rms}=\sqrt{\frac{1}{n}\overset{n}{\underset{i=1}{\sum }}{\left(T_{oral}-T_{ref}\right)}^{2}}$, where *n* is the number of subjects) between $T_{oral}$ and $T_{ref}$, is another metric used to assess clinical measurement accuracy in medical devices [51]. While $A_{rms}$ will not indicate the direction of error (e.g., overestimate or underestimate) and error distribution, it does quantify the cumulative magnitude of error. We implement it here to provide a single accuracy metric that combines the impact of bias and precision, as well as to ensure that positive and negative local bias values do not cancel out to give an erroneous impression of strong performance, as can occur with $\Delta_{cb}$.

Regression analysis [50] can also provide useful insight into the quality of temperature measurements. We generated scatter plots of $T_{oral}$ against $T_{ref}$ and fit linear trendlines to the data; these curves were then compared with the ideal (i.e., $T_{oral}$ = $T_{ref}$). Pearson correlation coefficients (r values) were also obtained to quantify the degree of linear correlation between $T_{oral}$ and $T_{ref}$.

#### 2.6.2. Metrics for Diagnostic Performance

In addition to methods focused on temperature measurement accuracy, we also implemented diagnostic performance assessment techniques to evaluate fever screening effectiveness for each IRT. These analyses involved calculation of sensitivity (true positive rate, *Se* = *TP*/*P*, where *TP* and *P* represent true positive and condition positive respectively) and specificity (true negative rate, *Sp* = *TN*/*N*, where *TN* and *N* represent true negative and condition negative respectively). The focus of this approach is to determine whether febrile subjects can be detected given specific reference temperature thresholds ($T_{thresh}$). The value for $T_{thresh}$ was set to 37.5 °C to define P ($T_{ref}\;$ > $T_{thresh}$) and N ($T_{ref}\;$ < $T_{thresh}$) for fever screening [2,27]. We also defined a cutoff temperature ($T_{cut}$) to determine positive or negative results based on $T_{oral}$. Based on the P, N, predicted P ($T_{oral}\;$ > $\;T_{cut}$) and predicted N ($T_{oral}\;$ < $\;T_{cut}$) for all subjects, TP ($T_{oral}\;$ > $\;T_{cut}$ and $T_{ref}\;$ > $T_{thresh}$) and TN ($T_{oral}\;$ < $\;T_{cut}$ and $T_{ref}\;$ < $T_{thresh}$) were obtained to calculate *Se* and *Sp*. At each $T_{cut}$, a pair of *Se*/*Sp* values were determined. An ROC curve for each facial temperature location was generated from 1000 $T_{cut}$ values equally spaced between 30 °C and 40 °C. The area under the ROC curve (AUC), an effective and combined measure of *Se* and *Sp*, was calculated to provide an aggregate measure of performance, where a maximum AUC of 1 indicates perfect diagnostic performance in differentiating diseased with non-diseased subjects [52,53]. The value of $\sqrt{{\left(1-Se\right)}^{2}+{\left(1-Sp\right)}^{2}}$, notated as $d_{SeSp}$, indicates the distance between the coordinate points of (1 − *Sp*, *Se*) and (0, 1), the perfect 1 − *Sp* and *Se* values [52]. The smaller the $d_{SeSp}$ value, the better the performance. The value of $d_{SeSp}$ at $T_{cut}\;$ = $T_{thresh\;}$ = 37.5 °C was used to evaluate the fever screening performance.

## 3. Results

### 3.1. Regression Methods for Calibration

As mentioned in Section 2.5, the training data (for 612 and 606 subjects with IRT-1 and IRT-2 respectively) were used to determine the relationship between different $T_{skin}$ ($T_{FC}$, $T_{FCmax}$, $T_{CEmax}$ or $T_{max}$) and $T_{ref}$ with different regression methods (constant offset, ordinary linear, quadratic, and Deming). We also implemented weighted linear, binning, and segmented linear regression methods due to the nonuniform distribution of temperatures. While the quadratic method usually showed nearly identical regression curves (Figure 3) with the segmented linear regression method, it led to nonmonotonic regression curves for some cases. Therefore, only the segmented linear regression method is discussed in this paper.

Figure 4 shows regression curves based on the training data. The segmented linear regression curve is omitted for simplification in this figure. We used different $T_{skin}$ as independent variables (*x*-axis) and $T_{ref}$ as the dependent variable (*y*-axis) in all the regression methods. In Section 4.1, we will briefly discuss the methods of using $T_{ref}$ as independent variable.

The results in Figure 4 indicate that lines for constant offset, ordinary linear, and Deming regression methods exhibit a common point of concurrency in each graph, near $T_{ref}\;$ ≈ 37 °C, $T_{FC}$ ≈ 34.5 °C, $T_{FCmax\;}$ ≈ 35 °C, $T_{CEmax}\;$ ≈ 35.5 °C, and $T_{max}\;$ ≈ 35.7 °C for both IRT-1 and IRT-2. That these lines intersect near a single point is likely because the least squares approach minimizes the sum of squared residuals, which means each data point contributes equally to the sum. Therefore, a temperature interval with more data will have larger impact on the fitting equation. The location of each point of concurrency is related to the mean temperature offset between the reference value and facial measurements, which was discussed previously [41]. Figure 5 shows the kernel density curves of $T_{ref}$, $T_{FC}$, $T_{FCmax}$, $T_{CEmax},$ and $T_{max}$ for IRT-1 and IRT-2. The curves for both IRTs are very similar, with the peak density for each site matching the corresponding points of concurrency. The Pearson correlation coefficients between $T_{ref}$ and $T_{FC}$/$T_{FCmax}$/$T_{CEmax}$/$T_{max}$ for IRT-1 are 0.53, 0.60, 0.79 and 0.82 respectively. These numbers for IRT-2 are 0.52, 0.57, 0.80, and 0.82.

### 3.2. Temperature Measurement Accuracy—Quantitative Analysis

The testing data (for 408 and 404 subjects with IRT-1 and IRT-2, respectively) were used to evaluate temperature measurement accuracy. The calibration curves based on different regression methods were applied to impute $T_{oral}$ from different $T_{skin}$ values ($T_{FC}$, $T_{FCmax}$, $T_{CEmax}$ or $T_{max}$). By comparing final imputed $T_{oral}$ with $T_{ref}$, temperature measurement accuracy could be evaluated in different ways, as described in Section 2.6.

To calculate clinical bias ($\Delta_{cb}$), clinical bias SD ($\sigma_{\Delta cb}$), and root-mean-square difference ($A_{rms}$), we separated the testing data into three intervals based on $T_{ref}$: $T_{ref}$ < 37 °C, 37 °C ≤ $T_{ref}$ ≤ 38.5 °C, and $T_{ref}$ > 38.5 °C. Since the diagnostic threshold ($T_{thresh}$, the $T_{ref}$ to define condition positive/negative) for fever screening is usually between 37.5 and 38 °C [41], the interval of 37.0–38.5 °C is particularly important. Results for $\Delta_{cb}$, $\sigma_{\Delta cb},$ and $A_{rms}$ were calculated for the entire testing set and each of the three intervals. As described in our previous study (Figure 2 in [41]), we acquired thermal images of each subject in four rounds. During each round of imaging, each IRT acquired three consecutive frames (acquisition time ~0.1 s) that were averaged to reduce noise and form a single thermal image. All analysis in this article was based on the averaged thermal images from the first round of measurements, except for the clinical repeatability ($\sigma_{r}$) analysis. To calculate $\sigma_{r}$, the SD of three $T_{oral}$ temperatures based on the averaged thermal images from each of the first three rounds of measurements was calculated for each subject and then pooled based on the ISO 80601-2-56 standard [34].

Table 5 and Table 6 display key metrics ($\Delta_{cb}$, $\sigma_{\Delta cb}$, $A_{rms}$, and $\sigma_{r}$) for $T_{CEmax}$- and $T_{max}$-based $T_{oral}$ for IRT-1 and IRT-2 respectively. In these results, the minimum $\Delta_{cb}$, $\sigma_{\Delta cb}$ and $A_{rms}$ values for all subjects and subjects with $T_{ref}$ < 37 °C generally come from the segmented linear regression method for both IRTs. The smallest $\Delta_{cb}$ values over the range 37 °C ≤ $T_{ref}$ ≤ 38.5 °C are between ±0.1 °C for both IRTs, coming from the constant offset, weighted linear, and binning methods. The related $\sigma_{\Delta cb}$ and $A_{rms}$ values over this range are less than 0.4 °C. The average $\sigma_{r}$ for both IRTs and all regression methods is 0.14 °C, with the minimum and maximum values of 0.07 °C and 0.23 °C. There is no one regression method that can achieve the best values for all the metrics and both IRTs. Later, we will demonstrate that temperature measurement accuracy over the range 37 °C ≤ $T_{ref}$ ≤ 38.5 °C is more related to diagnostic performance.

### 3.3. Temperature Measurement Accuracy—Graphical Analysis

Results that characterize variations in IRT temperature measurement accuracy are displayed graphically to elucidate variations across the covered temperature range and the presence of exceptional values or outliers. Scatter and difference plots provide useful tools for these types of analyses.

#### 3.3.1. Scatter Plots

A scatter plot provides a direct qualitative illustration of the clinical accuracy and the underlying variability of the relationship between $T_{oral}$ and $T_{ref}$. In the plots, we used $T_{ref}$ as the *x*-axis and $T_{oral}$ imputed from different $T_{skin}$ values as the *y*-axis. Figure 6 shows example scatter plots of $T_{oral}$ imputed from $T_{max}$ based on the constant offset, weighted linear, binning, and segmented linear regression methods versus $T_{ref}$ for IRT-1, since these methods show at least one of the best performance metrics in Table 5 and Table 6. Plots for $T_{oral}$ imputed from other $T_{skin}$, based on other regression methods, and for IRT-2 are not presented here due to space limitations.

Results in Figure 6 indicate that the segmented method produced the best fit (largest R^2^ value), whereas the binning method produced the trend line that was closest to the ideal $T_{oral}$ = $T_{ref}$ line. Given the highly non-uniform distribution of data, small differences in the slopes of the trend lines do not reflect overall accuracy differences. Two vertical lines at $T_{ref}$ = 37 °C and 38.5 °C separate the data into three temperature intervals for comparison with Table 5. Data above the ideal trend line cause a positive $\Delta_{cb}$ and vice versa. A wide data distribution in the vertical direction correlated with a large $\sigma_{\Delta cb}$. For example, the points in Figure 6c are the most dispersed in the vertical direction although the trend line is close to the ideal line, and the points in Figure 6d are the least dispersed. This indicates that $\sigma_{\Delta cb}$ for the binning method is the largest and $\sigma_{\Delta cb}$ for the segmented linear method is the smallest among the four regression methods, as have been shown in Table 5. Therefore, the trend line slope and intercept, the data point variability, and the coefficient of determination should be considered all together when reading a scatter plot. A direct qualitative view of the clinical accuracy through a scatter plot should be supported by quantitative values of other metrics, such as $\Delta_{cb}$, $\sigma_{\Delta cb}$, $A_{rms}$, $\sigma_{r}$, and *Se*/*Sp*/$d_{SeSp}$.

#### 3.3.2. Difference Plots

A difference plot directly shows the distribution of all the data that are used to calculate $\Delta_{cb}$ and $\sigma_{\Delta cb}$. It can also be used to identify proportional bias. The vertical axis of the plot is the difference between $T_{oral}$ and $T_{ref}$. The horizontal axis is the average of $T_{oral}$ and $T_{ref}$. About 95% of the difference values will fall in the range of $\Delta_{cb}\;$ ± 2$\sigma_{\Delta cb}$ if the values are normally distributed [34]. The difference plots for $T_{oral}$ calculated from $T_{max}$ based on the constant offset, weighted linear, binning, and segmented linear regression methods for IRT-1 are displayed in Figure 7 as examples. The first impression from Figure 7 is that some plots have an apparent trend (proportional bias), which is also seen in the corresponding scatter plots in Section 3.3.1 and Appendix A. For example, $T_{oral}$ and $T_{ref}$ show strong correlation in Figure 6d, yet more $T_{oral}$ values tend to be higher than $T_{ref}$ at lower temperatures and lower than $T_{ref}$ at higher temperatures. A corresponding trend of proportional bias is seen in Figure 7d. On the other hand, a slight trend might still exist even if two sets of data have a high degree of agreement [54]. For the $T_{max}$-based $T_{oral}$, the segmented linear regression method provides the smallest $\Delta_{cb}$ and $\sigma_{\Delta cb}$ that agrees with Table 5.

### 3.4. Diagnostic Performance

Variations in the ability of IRT systems to detect febrile subjects were analyzed using the *Se*/*Sp* approach based on clinically relevant thresholds. The ROC curves based on $T_{oral}$ imputed from each $T_{skin}$ under different regression methods were generated (not shown in this paper to reduce space), from which the *Se*/*Sp* values for $T_{cut}$ = $T_{thresh}$ = 37.5 °C were derived and the $d_{SeSp}$ values were calculated. Table 7 shows the *Se*/*Sp* and $d_{SeSp}$ values for $T_{CEmax}$- and $T_{max}$-based $T_{oral}$ with different regression methods. Compared with Table 5 and Table 6, we can see a strong relationship between $\Delta_{cb}$/$\sigma_{\Delta cb}$/$A_{rms}$ values in the range of 37 °C ≤ $T_{ref}$ ≤ 38.5 °C and *Se*/*Sp*—the minimum values of $\Delta_{cb}$/$\sigma_{\Delta cb}$/$A_{rms}$ are correlated to the minimum values of $d_{SeSp}$ (i.e., the largest *Se*/*Sp* combination). The smallest $\Delta_{cb}$/$\sigma_{\Delta cb}$/$A_{rms}$ values over the range 37 °C ≤ $T_{ref}$ ≤ 38.5 °C (Table 5 and Table 6), as well as optimum *Se*/*Sp* combinations for $T_{oral}$ (Table 7) come from the constant offset, weighted linear, and binning methods. On the other hand, the temperature measurement metrics over the full temperature range are not related to the $d_{SeSp}$ values. Therefore, if an IRT is designed for fever screening, the clinical accuracy in the range of 37–38.5 °C (oral cavity as the reference site) is more important than in other ranges. An IRT with the smallest $\Delta_{cb}$/$\sigma_{\Delta cb}$/$A_{rms}$ values within the whole temperature range does not necessarily mean it has the best *Se*/*Sp* for fever screening. For example, the *Se*/*Sp* values based on the segmented regression method are the worst for $T_{CEmax}$- and $T_{max}$-based $T_{oral}$ due to the large $\Delta_{cb}$ values in the range of 37.0 °C ≤ $T_{ref}$ ≤ 38.5 °C, although the values of $\Delta_{cb}$, $\sigma_{\Delta cb}$ and $A_{rms}$ based on this method across the full temperature range are the best.

To further analyze this issue, we defined the optimal cutoff temperature ($T_{op.cut}$) as the $T_{cut}$ that minimizes $d_{SeSp}$ (lengths of green line segments in Figure 8) [52], as obtained from the ROC curve. We also define predicted optimal cutoff temperature ($T_{p.op.cut}$) as the $T_{cut}$ imputed based on $T_{thresh}$ and $\Delta_{cb}$ in the temperature range of 37.0–38.5 °C, $T_{p.op.cut}\;$= $T_{thresh\;}$ + $\Delta_{cb}$. For brevity, we only show the ROC curves based on $T_{oral}$ imputed from $T_{max}$ and regression methods of constant offset, weighted linear, and segmented linear for IRT-1 in Figure 8. The *Se*/*Sp* values for $T_{cut}$ equals $T_{op.cut}$, $T_{p.op.cut},$ and $T_{thresh}$ are labeled together in each graph. From Figure 8, the $T_{op.cut}$ and $T_{p.op.cut}$ values are rather close with a difference of less than 0.1 °C, except for the segmented linear graph with a difference of 0.16 °C. The average difference between $T_{op.cut}$ and $T_{p.op.cut}$ is as small as 0.08 °C. The results indicate that the fever screening performance of an IRT can be optimized by adjusting the $T_{cut}$ value based on $\Delta_{cb}$ in the range of 37 °C ≤ $T_{ref}$ ≤ 38.5 °C. Figure 8c also illustrates the poor Se values based on the segmented linear regression method in Table 5 because of large $\Delta_{cb}$ in the range of 37 °C ≤ $T_{ref}$ ≤ 38.5 °C.

### 3.5. Clinical Accuracy—IRTs Versus NCITs

There have been inconsistent conclusions regarding the clinical accuracy of IRTs versus NCITs. A document from the Centers for Disease Control and Prevention indicates that IRTs are not as accurate as NCITs and may be more difficult to use effectively [55]. However, several scientific studies have shown different opinions [21,22]. Further discussion of this topic is needed. As described in our previous article [41], the temperature of each subject was measured with two IRTs and six NCITs. A full analysis of the NCIT data is presented elsewhere [45]. Therefore, it is potentially useful to directly compare the clinical data collected by these two different IRTs and six models of NCITs. On the other hand, IRTs can measure temperature from different facial locations. The measurements from the forehead can be a surrogate for NCIT measurements and thus be used to indirectly compare NCIT and IRT performance.

#### 3.5.1. Direct Performance Comparison

During our clinical study, two different IRTs and six models of NCITs were used to collect temperature data from each subject. The laboratory and clinical accuracy of these six models of NCITs has been analyzed in references [56] and [45] respectively. Laboratory results indicate that five of the six NCIT models did not meet the laboratory acceptance criterion of ±0.3 °C recommended by the ASTM E1965-98:2016 standard [33]. The algorithms used by these NCITs to convert temperature from the measurement site to the reference site (i.e., regression methods for imputing $T_{oral}$ from $T_{skin}$) are unknown.

Clinical NCIT results (Table 2 in [45]) show that mean $\Delta_{cb}\;$± $\sigma_{\Delta cb}$ values for the six models (A, B, C, D, E, F) over the full temperature range were −0.26 ± 0.46 °C, −0.23 ± 0.42 °C, 0.15 ± 0.41 °C, −0.32 ± 0.58 °C, −0.88 ± 0.54 °C, and 0.22 ± 0.46 °C. Depending upon the NCIT model, 48–88% of the temperature measurements were beyond the labeled accuracy, which aligns well with the results from another study [57]. On the other hand, the worst/best $\Delta_{cb}\;$± $\sigma_{\Delta cb}$ values for $T_{max}$-based $T_{oral}$ across the full temperature range were −0.09 ± 0.41 °C/−0.03 ± 0.29 °C for IRT-1 and 0.19 ± 0.32 °C/0.01 ± 0.27 °C for IRT-2 (Table 5 and Table 6). These results indicate that the two IRTs have similar accuracy, and both have better bias and precision than the six models of NCITs, even with the worst regression method.

NCIT results (Figure 4 in [45]) also showed that for a $T_{thresh}$ of 37.5 °C, the *Se*/*Sp* values for the six models were 0.11/1.00, 0.35/0.99, 0.58/0.97, 0.40/0.98, 0.03/1.00, and 0.70/0.85 respectively, with the $d_{SeSp}$ values being 0.89, 0.65, 0.42, 0.60, 0.97, and 0.34, respectively. On the other hand, the *Se*/*Sp* values were 0.89/0.87 and 0.88/0.88 for $T_{CEmax}$- and $T_{max}$-based $T_{oral}$ measurements by IRT-1 calibrated with the weighted linear regression method, with the related $d_{SeSp}$ values being 0.18 and 0.17, respectively (Table 5 and Table 6). A comparison of these data indicates that IRTs can be more effective to screen subjects with EBT than NCITs.

#### 3.5.2. Indirect Comparison Based on Imaging Results

Given the similarities in physical working mechanism and facial location, IRT data for $T_{oral}$ calculated from $T_{FC}$ and $T_{FCmax}$ (Table A1 and Table A2 for IRT-1 and IRT-2, provided in Appendix A for brevity) may provide a useful surrogate for NCIT measurements. These results were compared with IRT data for $T_{oral}$ calculated from $T_{CEmax}$ and $T_{max}\;$(Table 5 and Table 6 for IRT-1 and IRT-2). From Table A1 and Table 5, the optimal $\Delta_{cb}$ and $\sigma_{\Delta cb}$ values across the full $T_{ref}$ range for $T_{CEmax}$- and $T_{max}$-based $T_{oral}$ have minimal differences from the values for $T_{FC}$- and $T_{FCmax}$-based $T_{oral}$. However, these values in the $T_{ref}$ range of 37–38.5 °C are 0.22 ± 0.35 °C and 0.18 ± 0.34 °C for $T_{FC}$- and $T_{FCmax}$-based $T_{oral}$ versus 0.05 ± 0.30 °C and 0.08 ± 0.29 °C for $T_{CEmax}$- and $T_{max}$-based $T_{oral}$ respectively. Multiple comparisons were performed between the four sets of $\Delta_{cb}$ values (noted as A, B, C and D) for $T_{FC}$-, $T_{FCmax}$-, $T_{CEmax}$- and $T_{max}$-based $T_{oral}$ data using the Tukey Honest Significant Difference method. The results indicate that the forehead measurement site typically used by NCITs tends to provide poorer accuracy than a full-face approach or one that targets the inner canthus (*p*-values < 0.05 between A/B and C/D). On the other hand, there is no significant difference between A and B or C and D (*p*-values > 0.05), indicating the full-face and inner cantus approaches have similar optimal $\Delta_{cb}$ and $\sigma_{\Delta cb}$ values.

Comparisons of diagnostic performance for EBT detection between these measurement approaches can also be made from data in Table 7, Table A1 and Table A2. The optimal *Se*/*Sp* values identified for IRT-1 are 0.67/0.82 or 0.74/0.72 for $T_{FC}$-based $T_{oral}$, 0.67/0.87 or 0.72/0.78 for $T_{FCmax}$-based $T_{oral}$ (Table A1), versus 0.89/0.87 for $T_{CEmax}$-based $T_{oral}$, and 0.88/0.89 for $T_{max}$-based $T_{oral}$ (Table 7). The results for IRT-2 in Table 7 and Table A2 are similar. The optimal $d_{SeSp}$ values identified for both IRTs are between 0.31 and 0.38 for $T_{FC}$- and $T_{FCmax}$-based $T_{oral}$, which are close to the best $d_{SeSp}$ value for the six models of NCITs.

Corresponding scatter plots, difference plots, and ROC curves based on $T_{oral}$ calculated from $T_{FC}$ are provided (Figure 1, Figure 2 and Figure 3 in Appendix A) for IRT-1 to mirror the results in Figure 6, Figure 7 and Figure 8, for $T_{oral}$ calculated from $T_{max}$. The ROC curves for $T_{FC}$ are significantly lower than the curves for $T_{max}$, which agree with the *Se*/*Sp* values in Table 7 and Table A1 and indicate the potential low *Se*/*Sp* values of NCITs. The scatter plots of $T_{FC}$-based $T_{oral}$ versus $T_{ref}$ (Figure A1) are more dispersed and their trend lines are further from the ideal line than the graphs for $T_{max}$, indicating larger $\Delta_{cb}$ and $\sigma_{\Delta cb}$ for $T_{FC}$-based $T_{oral}$. Comparisons of difference plots for $T_{FC}$- and $T_{max}$-based $T_{oral}$ show the same conclusion.

## 4. Discussion

Through an extensive clinical study of over 1000 subjects, we have evaluated the clinical accuracy of two IRTs under controlled conditions for temperature measurement. The clinical accuracy of the IRTs has been quantitatively evaluated with different metrics including $\Delta_{cb}$, $\sigma_{\Delta cb}$, $A_{rms}$, $\sigma_{r}$, and *Se*/*Sp*/$d_{SeSp}$. Dividing the data into training and testing sets, we have studied the impact of calibration approaches and methods for establishing diagnostic cutoff temperatures, and elucidated differences in performance between IRTs and NCITs. The results are displayed with scatter plots, difference plots and ROC curves. Overall, these findings provide unique and valuable insights into both the optimization and assessment of IRT-based devices for temperature estimation and fever detection.

### 4.1. Effects of Regression Methods on the Clinical Accuracy

Our analysis of regression approaches indicated no clear optimal method that can improve all clinical accuracy metrics. A specific regression method tended to provide the best clinical accuracy in terms of a specific metric. When the full range of temperatures were considered in our data, the segmented linear regression provided the smallest $A_{rms}$ values, the least scatter (and the highest *R*^2^ value) in Figure 6, and the narrowest difference distribution range in Figure 7. However, when we restricted the temperature range to the diagnostic zone (37 °C ≤ $T_{ref}$ ≤ 38.5 °C), the constant offset, weighted linear, and binning methods provided the highest *Se*/*Sp* and the smallest bias.

To apply different regression methods to find the relation between $T_{skin}$ and $T_{ref}$, we used $T_{skin}$ and $T_{ref}$ as independent and dependent variables, respectively. In theory, the independent variable should be the one that is more accurate, in our case, $T_{ref}$. If we used $T_{ref}$ and $T_{skin}$ as independent and dependent variables respectively, the function we obtained will be $T_{skin}$ = *f*($T_{ref}$). During the evaluation, this function should be used inversely ($T_{oral}=f{\;}^{-1}\left(T_{skin}\right)$) to convert $T_{skin}$ to $T_{oral}$. The inverse operation might cause extra errors. We applied the inverse equations of these regression equations to the testing data and calculated the same clinical accuracy metrics (For brevity, not included in this paper) as shown in Table 5, Table 6 and Table 7. We did not find clinical accuracy improvement in terms of these metrics.

### 4.2. Metrics and Requirements for Evaluating Clinical Accuracy

Table 5, Table 6 and Table 7 show different clinical accuracy metrics for IRT-1 and IRT-2 respectively, including $\Delta_{cb}$, $\sigma_{\Delta cb}$, $A_{rms}$, $\sigma_{r}$, and *Se*/*Sp*/$d_{SeSp}$. While $\Delta_{cb}$ and $\sigma_{\Delta cb}$ are recommended in international thermometer standards, they do not necessary represent the optimal metrics for all applications. One limitation of $\Delta_{cb}$ as a performance metric is that it is mean value only reflecting the systematic bias and that large positive and negative local biases may cancel out, thus producing a small $\Delta_{cb}$ value, as if the local biases were small. Therefore, $\Delta_{cb}$ and $\sigma_{\Delta cb}$ should always be evaluated together. The metric $A_{rms}$ is the root-mean-square difference between measured values ($T_{oral}$) and reference values ($T_{ref}$) [51]. Being a single accuracy metric that combines the impact of $\Delta_{cb}$ and $\sigma_{\Delta cb}$, it helps ensure that positive and negative local bias values do not cancel out to give an erroneous impression of strong performance, as can occur with $\Delta_{cb}$. However, $A_{rms}$ does not indicate whether errors are mainly positive or negative and does not distinguish systematic and random errors. Another metric that was not discussed in this article, mean absolute error (MAE), is similar to $A_{rms}$ and might also be considered.

The values of $\Delta_{cb}$, $\sigma_{\Delta cb}$ and $A_{rms}$ for different temperature ranges might have different significance. If an IRT is designed for fever screening, then values of these metrics within the reference temperature range of 37–38.5 °C are more important than those based on the full temperature range, since they most directly impact diagnostic ability. For such a device, *Se*/*Sp* values for common $T_{thresh}$ values (e.g., 37.5 °C or 38 °C) might be stronger performance metrics than $\Delta_{cb}$ and $\sigma_{\Delta cb}$. The AUC value is commonly quoted for ROC curves [41], which may be a better metric for overall performance since it is an aggregate measure of diagnostic capability. The higher the AUC, the greater the potential of an IRT to distinguish subjects with and without EBT. To achieve the full potential of the IRT, the optimal cutoff temperature to obtain the least $d_{SeSp}$ can be predicted based on $T_{thresh}$ and $\Delta_{cb}$ in the temperature range of 37.0–38.5 °C, $T_{p.op.cut}\;$= $T_{thresh}$ + $\Delta_{cb}$. In reality, users can also increase or decrease $T_{cut}$ to increase Sp or Se at the cost of decreasing Se or Sp at the same time.

Relatively little consensus has been achieved in the establishment of minimum performance requirements for IRTs. Currently, we are only aware of one consensus requirement for IRT laboratory accuracy. The IEC 80601-2-59: 2017 standard [30] requires that laboratory error of IRTs be below 0.5 °C in the $T_{skin}$ range of 34–39 °C [32]. Performance requirements in thermometer standards may also be adapted for use with IRTs: ISO 80601-2-56:2017 for clinical thermometers [34], ASTM E1112-00:2011 for electronic thermometers [58], and ASTM E1965-98:2016 for infrared thermometers [33]. The maximum permissible errors defined in these standards are listed in Table 8.

None of the aforementioned standards includes clinical accuracy requirements for IRTs or thermometers. The ISO 80601-2-56:2017 standard provides a clinical example where $\Delta_{cb}\;$ ± $\sigma_{\Delta cb}$ is 0.07 ± 0.22 °C. The text indicates that the $\Delta_{cb}$ value is acceptable and the $\sigma_{\Delta cb}$ value could be considered by some to be clinically acceptable, although it is relatively high. The ASTM E1965-98:2016 standard also provides an example of clinical accuracy evaluation results for an infrared thermometer, with $\Delta_{cb}\;$ ± $\sigma_{\Delta cb}$ values of −0.25 ± 0.35 °C, −0.16 ± 0.18 °C, and 0.11 ± 0.21 °C for age groups of infants, children, and adults, respectively. The standard indicates that the thermometer under test may not be sufficiently accurate for use on infants since errors in temperature measurements may be clinically significant. Nevertheless, these examples do not define clinical accuracy requirements. Based on our study, an IRT can provide a good fever screening performance ($d_{SeSp}\;$ ≤ 0.2) if $\sigma_{r\;}$ ≤ 0.2 °C and its temperature measurement accuracy satisfies these requirements within the temperature range of 37.0–38.5 °C with oral cavity as the reference body site: −0.1 °C ≤ $\Delta_{cb}$ ≤ 0.1 °C, $\sigma_{\Delta cb}$ ≤ 0.4 °C, $A_{rms}$ ≤ 0.4 °C. For our IRTs, these requirements are met for the $T_{CEmax}$- and $T_{max}$-based $T_{oral}$ data imputed with the weighted linear (for IRT-1 and IRT-2) and constant offset (for IRT-2 only) methods.

### 4.3. Difference Plot Methods

In Section 3.3.2, we used the mean of $T_{oral}$ and $T_{ref}$ as the horizontal axis of the difference plots, based on the Bland–Altman approach. In theory, the horizontal axis of the plot is determined based on the best estimate of the true values [50]. While we believe $T_{ref}$ is more accurate than $T_{oral}$, $T_{ref}$ also presents error with the SD of two measurements being ~0.1 °C. Moreover, there is no consensus in the literature as to the optimal approach for thermographic data analysis. Bland and Altman argued that the difference against the reference measurements will show a relationship between them when none exists [54]. Therefore, they recommended that the mean value be used on the horizontal axis. However, researchers still often use reference values alone as the horizontal axis [50,59,60], believing reference values are the best estimate of the true values. We redrew the difference plots of Figure 7 with $T_{ref}$ as the horizontal axis, as shown in Figure 9. From the figure, we can see that the trends in Figure 9 are different from the trends in Figure 7. Negative correlation can be seen in Figure 9 as Bland and Altman predicted [54]. However, a significant advantage of one approach over the other is not clearly apparent.

### 4.4. Performance Comparison of IRTs and NCITs

IRTs and NCITs represent the primary device types currently used in practice for real-time measurement of EBT during epidemics [17,18,19,29]. They both use passive remote sensing technologies that detect mid- and/or long-wave IR radiation and convert measurements to temperature based on the Stefan–Boltzmann law [61]. NCITs estimate temperature at a reference body site (usually oral) based on radiation from a small region of skin (e.g., forehead) [33], whereas IRTs provide a 2D temperature distribution of the face and may target a specific region (e.g., inner canthi) [30]. FDA has cleared NCITs to independently measure human body temperature, yet no IRT has been cleared for a similar purpose. Current IRTs on US market are only authorized for emergency use [62]. In several scientific studies, the accuracy of NCITs has been called into question, particularly relative to IRTs [21,22]. Our study provides another angle to compare IRTs with NCITs.

Both indirect and direct comparisons of IRTs with NCITs indicate that when designed for optimal performance, the clinical accuracy of IRTs will likely be greater than that of NCITs. The two IRTs have similar accuracy, and both have better bias and precision than the six models of NCITs, even with the worst regression method. One reason for this may be the use of the forehead as the NCIT measurement location. The skin temperature at this location tends to be sensitive to environmental factors such as ambient temperature and airflow, which may degrade correlation with core/oral temperature [23]. The IRTs implemented in the current study also use higher performance electronic components than the typical portable NCIT, and thus are much more expensive. Of course, in order for an IRT to achieve a high degree of clinical accuracy it will need to meet laboratory accuracy requirements [32], have an effective algorithm to convert the measured skin temperature to the temperature at a reference body site (e.g., oral cavity), and be deployed and operated according to established best practices.

In summary, from both temperature measurement accuracy and diagnostic performance standpoints, approaches based on forehead measurements, as with most NCITs, are likely to be inferior to those involving the full face or inner canthus measurements recommended for IRTs.

### 4.5. Study Challenges and Limitations

While our clinical study provided important insights, it is worth noting some of the key challenges we faced and the limitations to our findings. For example, the distribution of reference temperatures acquired is clearly uneven. Most subjects had oral temperatures of 37.0 ± 0.5 °C and the number of subjects with an EBT was limited. While the temperature distribution across a typical population would likely be somewhat Gaussian, an optimal data set would provide a more uniform distribution of temperatures across the normal through febrile range. However, it was difficult to recruit febrile subjects, which is a common problem for clinical fever screening studies [25]. Our study was initially designed to have a large population (~1000 subjects) in order to accrue a statistically significant sample of febrile subjects, despite a relatively low prevalence. As a result, we were able to obtain a greater number of data sets from febrile subjects than most clinical studies.

Perhaps the most significant caveat to our results is the limited age range of the study population. Overall, 95% of subjects were under 30 years of age. Research on the effect of age on IRT accuracy is limited, yet one paper has shown that the best correlation of IRT temperatures with core temperature is seen in children (aged 3–18 years) [63]. While our study did not include subjects below 18 years old, about half were in the 18–21 range. Therefore, the results in this paper might not represent the accuracy for all age groups. A clinical study for system validation should cover all age groups, dependent on the device application. Since the two sets of data for training and testing were based on the same pool of data and random selection was used to determine the two sets, the performance estimates may be biased (upwards) and not generalizable in the target population [64]. As such, it is likely that our study may represent a best-case scenario.

The subject circadian rhythm might also affect fever screening performance. For example, different studies have shown that core body temperature in the morning maybe 0.3–0.9 °C lower than in the afternoon [13,14,65]. We did not consider circadian rhythm in our analysis, yet additional study of this variable and the need for methods to mitigate its impact in infectious disease screening is warranted [66]. In the future, we intend to provide additional retrospective analysis of our data to assess this potential confounding factor.

To minimize the influence of outside temperature, a 15-min acclimation period was implemented prior to the start of measurements. However, oral temperature might still be affected by smoking or ingestion of cold or hot food or beverage during this time [67]. To mitigate this potential confounder, we extracted data sets for which the difference between the two oral temperature readings was greater than 0.5 °C as well as those where only one oral temperature reading was recorded. These exclusions amounted to 56 subjects. Such checks on data quality are useful for ensuring the validity of clinical IRT data [49].

## 5. Conclusions

Overall, our large-scale clinical study has generated unique and highly valuable quantitative information on fever-screening IRT performance and helped to identify potential best practices for the calibration and evaluation of IRT clinical accuracy. Current findings on IRT diagnostic performance were generally consistent with our prior analysis of results from 500 subjects, indicating IRTs have a strong potential for achieving high sensitivity and specificity in the detection of EBT. Algorithms used to impute oral cavity temperature based on skin temperature are critical for accurate clinical measurement. A simple offset approach may be effective in many situations, but when calibration data sets involve a high proportion of normal-range temperatures, then methods that account for this uneven distribution have key advantages. While metrics recommended in standards provide useful insights into IRT performance, implementing additional approaches like A_rms_ to assess temperature measurement accuracy and *Se*/*Sp* for clinical diagnostic accuracy may be beneficial. Moreover, temperature measurement accuracy within a temperature window near the diagnostic threshold for fever may be more important for evaluating fever screening IRTs than accuracy within a full temperature range.

Direct and indirect comparisons of our custom IRT systems with commercial NCITs showed that the former (i.e., IRT systems) were more accurate and provide greater diagnostic efficacy. Our results indicate that this is due at least partly to the fact that IRTs measure temperature from a more thermally stable facial location provided by a large number of pixels (e.g., 320 × 240 pixels). The superior capability of IRTs may enable the detection of lower grade and/or earlier stage fevers. Compared with NCITs, IRTs might be a better choice for fever screening in high-traffic areas or higher-risk locations where the higher cost could be justified by greater effectiveness. Furthermore, an IRT operator is not required to be in physical proximity to the subject (e.g., the distance between subject and IRTs was 0.6–0.8 m in this study). Indeed, they could even be in a different area or room, or a completely automated approach could be implemented, thus reducing the risk of infection. Another advantage of IRTs is their ability to provide temperature data from a range of facial locations, such as the inner canthi for fever detection [41]. Spatial variations in facial temperature can also be related to certain diseases (e.g., skin inflammatory conditions, breast cancer, systemic inflammatory diseases, septic shock, and the healing potential of wounds) [68]. Finally, it should be noted that additional study of our clinical results will be needed to elucidate additional confounding factors.

## Figures and Tables

**Figure 1 sensors-22-00215-f001:**
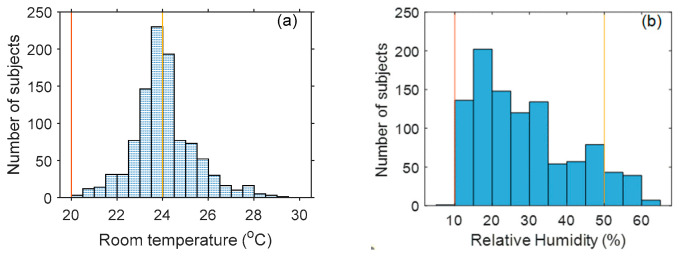
Ambient temperature and relative humidity histogram during the clinical study. (The range between the two vertical lines indicate ideal ambient temperature/humidity based on ISO/TR 13154:2017).

**Figure 2 sensors-22-00215-f002:**
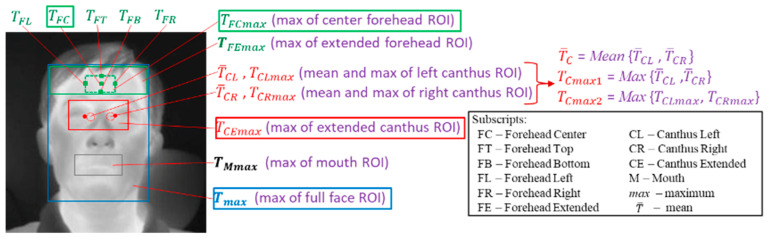
Delineated facial regions and critical points on thermal images [41].

**Figure 3 sensors-22-00215-f003:**
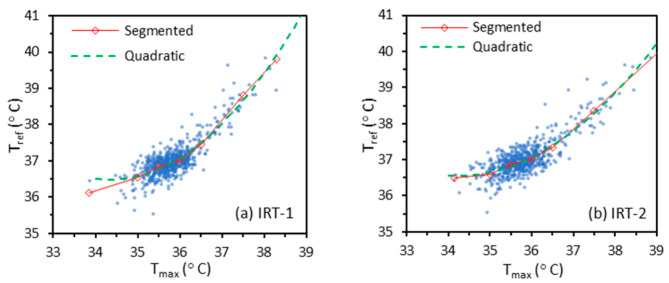
Examples of quadratic and segmented regression methods with $T_{max}$ and $T_{ref}$ as independent and dependent variables respectively for IRT-1 and IRT-2.

**Figure 4 sensors-22-00215-f004:**
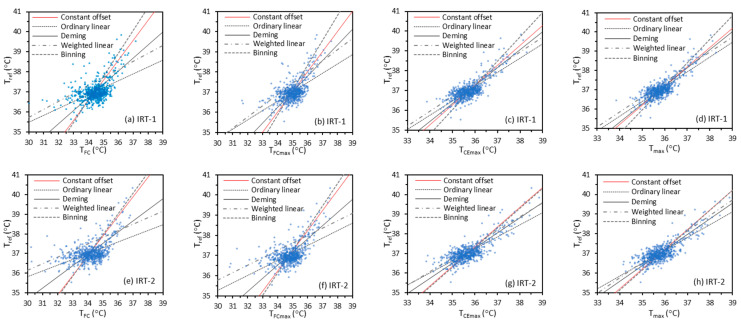
Different linear regression methods with $T_{skin}$ ($T_{FC}$, $T_{FCmax}$, $T_{CEmax}$, $T_{max}$ ) as independent variables and $T_{ref}$ as dependent variable for IRT-1 and IRT-2.

**Figure 5 sensors-22-00215-f005:**
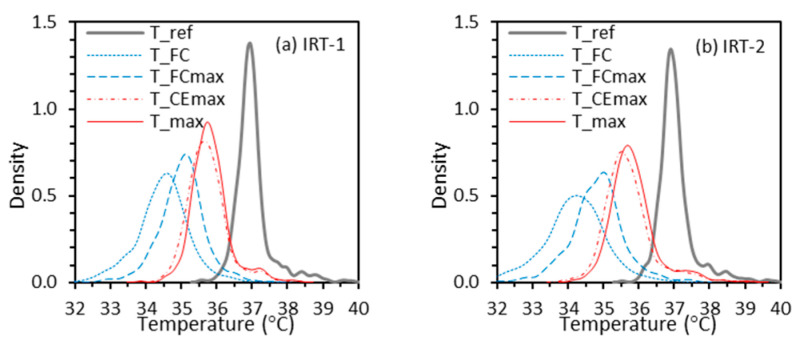
Kernel density curves to estimate the probability density functions of $T_{ref}$, $T_{FC}$, $T_{FCmax}$, $T_{CEmax}$ and $T_{max}$.

**Figure 6 sensors-22-00215-f006:**
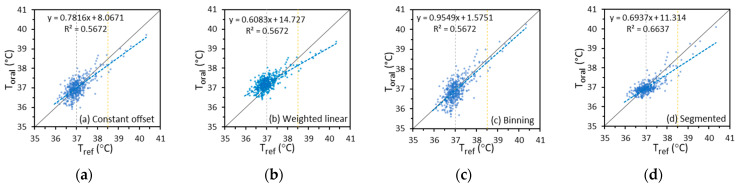
Scatter plots of $T_{oral}$ imputed from $T_{max}$ based on different regression methods versus $T_{ref}$ for IRT-1 (Dashed lines: trend lines of $T_{oral}$ versus $T_{ref}$; Solid lines: ideal trend lines of $T_{oral}$ = $T_{ref}$ ).

**Figure 7 sensors-22-00215-f007:**
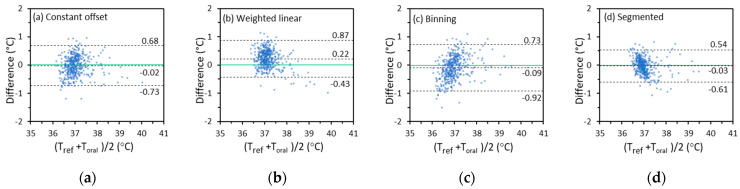
The temperature difference between $T_{max}$-based $T_{oral}$ and $T_{ref}$ versus their average for IRT-1 in the entire temperature range (Solid lines: lines of zero difference. Dashed lines: lines of difference being ∆_*cb*_ + 2σ_∆*cb*_, ∆_*cb*_, and ∆_*cb*_ − 2σ∆_*cb*_ respectively).

**Figure 8 sensors-22-00215-f008:**
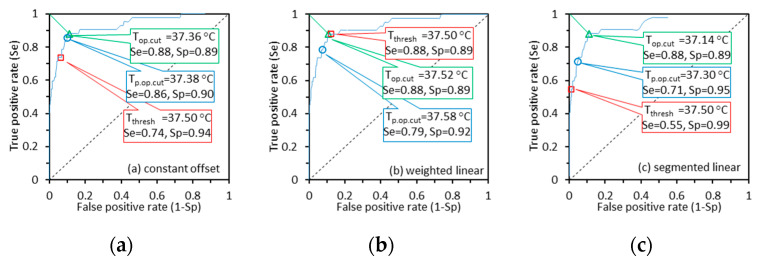
The ROC curves based on $T_{oral}$ imputed from $T_{max}$ and regression methods of constant offset, weighted linear and segmented linear for IRT-1. The triangle, circle and squre markers on curves show the *Se*/*Sp* values when $T_{cut}$ equals $T_{op.cut}$, $T_{p.op.cut}$, and $T_{thresh}$ respectively.

**Figure 9 sensors-22-00215-f009:**
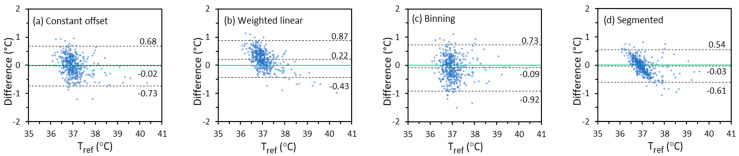
The temperature difference between $T_{max}$-based $T_{oral}$ and $T_{ref}$ versus $T_{ref}$ for IRT-1 in the entire temperature range (Solid lines: lines of zero difference. Dashed lines: lines of difference being ∆_*cb*_ + 2σ∆_*cb*_, ∆_*cb*_, and ∆_*cb*_ − 2σ∆_*cb*_ respectively).

**Table 1 sensors-22-00215-t001:** Demographics of study subjects.

		IRT-1		IRT-2	
		Subjects	%	Subjects	%
	Female	606	59.41	601	59.50
	Male	414	40.59	409	40.50
Age	18–20	534	52.35	527	52.18
	21–30	432	42.35	429	42.48
	31–40	31	3.04	31	3.07
	41–50	9	0.88	9	0.89
	51–60	11	1.08	11	1.09
	>60	3	0.29	3	0.30
Ethnicity	White	506	49.61	500	49.50
	Black/African-American	143	14.02	143	14.16
	Hispanic/Latino	57	5.59	55	5.45
	Asian	260	25.49	258	25.54
	Multiracial	50	4.90	50	4.95
	American Indian	4	0.39	4	0.40
$T_{ref}$ > 37.5 °C		111	10.88	111	10.99

**Table 2 sensors-22-00215-t002:** Study subject grouping by ambient temperature.

	Ambient Temperature (°C)	Relative Humidity	Subject # for IRT-1	Subject # for IRT-2
Group 1 [41]	20–24	10–62% (7.5% subject data in the 50–62% range)	544	540
Group 2	24–29	10–62% (9.9% subject data in the 50–62% range)	476	470

**Table 3 sensors-22-00215-t003:** Pearson correlation coefficients (*r* values) between facial temperatures and $T_{ref}$.

		Forehead							Inner Canthi								Mouth	Face
		$T_{FC}$	$T_{FT}$	$T_{FB}$	$T_{FL}$	$T_{FR}$	$T_{FCmax}$	$T_{FEmax}$	${\bar{T}}_{CL}$	${\bar{T}}_{CR}$	${\bar{T}}_{C}$	$T_{Cmax1}$	$T_{CLmax}$	$T_{CRmax}$	$T_{Cmax2}\;$	$T_{CEmax}$	$T_{Mmax}$	$T_{max}$
Group 1 [41]	IRT-1	0.46	0.41	0.49	0.47	0.43	0.55	0.63	0.60	0.58	0.63	0.65	0.70	0.71	0.73	0.75	0.60	**0.78**
	IRT-2	0.46	0.39	0.49	0.46	0.41	0.54	0.62	0.53	0.51	0.56	0.59	0.70	0.69	0.73	0.76	0.60	**0.79**
Group 2	IRT-1	0.50	0.37	0.52	0.46	0.43	0.56	0.60	0.62	0.61	0.65	0.66	0.74	0.75	0.77	0.79	0.69	**0.81**
	IRT-2	0.50	0.37	0.53	0.46	0.42	0.57	0.61	0.63	0.56	0.62	0.65	0.73	0.72	0.76	0.80	0.69	**0.82**

Note: Definitions of these facial temperatures can be found in Figure 2 and our previous paper [41]. The bold font shows the best results (the highest *r*).

**Table 4 sensors-22-00215-t004:** AUC values for ROC curves based on different facial temperatures.

		Forehead							Inner Canthi								Mouth	Face
		$T_{FC}$	$T_{FT}$	$T_{FB}$	$T_{FL}$	$T_{FR}$	$T_{FCmax}$	$T_{FEmax}$	${\bar{T}}_{CL}$	${\bar{T}}_{CR}$	${\bar{T}}_{C}$	$T_{Cmax1}$	$T_{CLmax}$	$T_{CRmax}$	$T_{Cmax2}\;$	$T_{CEmax}$	$T_{Mmax}$	$T_{max}$
Group 1 [41]	IRT-1	0.82	0.79	0.82	0.80	0.81	0.84	0.86	0.88	0.87	0.88	0.88	0.94	0.93	0.94	0.95	0.89	0.95
	IRT-2	0.82	0.79	0.82	0.79	0.79	0.84	0.87	0.91	0.87	0.90	0.92	0.95	0.93	0.94	0.95	0.88	0.97
Group 2	IRT-1	0.82	0.76	0.82	0.80	0.78	0.85	0.87	0.93	0.91	0.93	0.93	0.97	0.96	0.97	0.97	0.91	0.97
	IRT-2	0.82	0.76	0.82	0.78	0.79	0.84	0.85	0.94	0.88	0.92	0.94	0.96	0.94	0.97	0.97	0.90	0.97

**Table 5 sensors-22-00215-t005:** Clinical accuracy of $T_{oral}$ measurements for IRT-1 based on $T_{CEmax}$ and $T_{max}$: $\Delta_{cb}$, $\sigma_{\Delta cb}$, $A_{rms}$, and $\sigma_{r}$ (unit: °C).

Toral Based on TCEmax								Toral Based on Tmax					
		Offset	Ordinary	Deming	Weighted	Binning	Segmented	Offset	Ordinary	Deming	Weighted	Binning	Segmented
All	$\Delta_{cb}$	−0.03	−0.03	−0.03	0.21	−0.13	−0.03	−0.02	−0.02	−0.02	0.22	−0.09	−0.03
T_ref_	$\sigma_{\Delta cb}$	0.40	0.35	0.37	0.35	0.47	0.30	0.35	0.33	0.34	0.33	0.41	0.29
	$A_{rms}$	0.40	0.35	0.37	0.41	0.49	**0.30**	0.35	0.33	0.34	0.39	0.42	**0.29**
T_ref_<	$\Delta_{cb}$	**0.05**	0.11	0.07	0.34	−0.10	0.10	**0.05**	0.10	0.07	0.34	−0.06	0.10
37 °C	$\sigma_{\Delta cb}$	0.37	0.29	0.34	0.29	0.45	0.22	0.33	0.27	0.32	0.27	0.40	0.21
	$A_{rms}$	0.38	0.30	0.35	0.45	0.46	**0.24**	0.34	0.29	0.32	0.44	0.41	**0.23**
37°C≤	$\Delta_{cb}$	−0.14	−0.19	−0.16	**0.05**	−0.19	−0.21	−0.12	−0.17	−0.13	**0.08**	−0.14	−0.20
T_ref_	$\sigma_{\Delta cb}$	0.40	0.30	0.36	0.30	0.50	0.30	0.35	0.28	0.33	0.29	0.43	0.28
≤38.5 °C	$A_{rms}$	0.42	0.35	0.39	**0.31**	0.53	0.37	0.37	0.33	0.35	**0.30**	0.45	0.35
T_ref_>	$\Delta_{cb}$	−0.42	−0.91	−0.58	−0.62	**−0.12**	−0.39	−0.49	−0.87	−0.58	−0.61	**−0.18**	−0.39
38.5 °C	$\sigma_{\Delta cb}$	0.26	0.24	0.24	0.23	0.36	0.35	0.23	0.22	0.22	0.22	0.31	0.36
	$A_{rms}$	0.48	0.93	0.62	0.65	**0.36**	0.51	0.53	0.90	0.62	0.65	**0.34**	0.52
$\sigma_{r}$		0.11	0.08	0.10	0.09	0.14	**0.07**	0.18	0.14	0.17	0.14	0.22	**0.13**

Note: The bold font shows the best results (i.e., minimum values of Δcb, σΔcb, Arms, and σr).

**Table 6 sensors-22-00215-t006:** Clinical accuracy of $T_{oral}$ measurement for IRT-2 based on $T_{CEmax}$ and $T_{max}$: $\Delta_{cb}$, $\sigma_{\Delta cb}$, $A_{rms}$, and $\sigma_{r}$ (unit: °C).

Toral Based on TCEmax								Toral Based on Tmax					
		Offset	Ordinary	Deming	Weighted	Binning	Segmented	Offset	Ordinary	Deming	Weighted	Binning	Segmented
All	$\Delta_{cb}$	0.02	0.03	0.03	0.25	−0.03	**0.02**	0.01	0.02	0.02	0.19	−0.04	**0.01**
T_ref_	$\sigma_{\Delta cb}$	0.42	0.32	0.35	0.33	0.42	**0.29**	0.38	0.31	0.32	0.32	0.39	**0.27**
	$A_{rms}$	0.42	0.32	0.35	0.41	0.42	**0.29**	0.38	0.31	0.32	0.37	0.39	**0.27**
T_ref_<	$\Delta_{cb}$	0.06	0.15	0.11	0.35	0.00	**0.15**	0.05	0.14	0.10	0.27	−0.01	**0.14**
37 °C	$\sigma_{\Delta cb}$	0.44	0.29	0.35	0.32	0.44	**0.23**	0.39	0.27	0.32	0.32	0.40	**0.22**
	$A_{rms}$	0.44	0.33	0.37	0.47	0.44	**0.27**	0.40	0.30	0.34	0.42	0.40	**0.26**
37 °C≤	$\Delta_{cb}$	**−0.05**	−0.14	−0.10	0.10	**−0.10**	−0.20	**−0.05**	−0.14	−0.10	**0.06**	−0.10	−0.19
T_ref_	$\sigma_{\Delta cb}$	0.38	0.26	0.30	0.28	0.38	0.23	0.35	0.25	0.28	0.28	0.35	0.22
≤38.5 °C	$A_{rms}$	0.38	0.29	0.31	**0.29**	0.40	0.30	0.35	0.28	0.30	**0.28**	0.37	0.29
T_ref_>	$\Delta_{cb}$	0.25	−0.58	−0.25	−0.17	0.21	**−0.09**	0.14	−0.57	−0.28	−0.13	**0.11**	−0.19
38.5 °C	$\sigma_{\Delta cb}$	0.39	0.22	0.28	0.25	0.39	0.47	0.36	0.21	0.27	0.26	0.37	0.38
	$A_{rms}$	0.44	0.62	0.36	**0.29**	0.42	0.45	0.36	0.61	0.38	**0.28**	0.36	0.41
$\sigma_{r}$		0.15	0.09	0.11	0.10	0.15	**0.07**	0.22	0.15	0.18	0.18	0.23	**0.12**

Note: The bold font shows the best results (i.e., minimum values of Δcb, σΔcb, Arms, and σr).

**Table 7 sensors-22-00215-t007:** Diagnostic accuracy of IRT-1 and IRT-2 based on $T_{oral}$ imputed from $T_{CEmax}$ and $T_{max}$: *Se*/*Sp* and $d_{SeSp}$.

		$T_{oral}$ $\;\mathrm{Based}\;\mathrm{on}\;T_{CEmax}$						$T_{oral}$ $\;\mathrm{Based}\;\mathrm{on}\;T_{max}$					
		Offset	Ordinary	Deming	Weighted	Binning	Segmented	Offset	Ordinary	Deming	Weighted	Binning	Segmented
	Se	0.73	0.61	0.73	0.89	0.73	0.61	0.74	0.60	0.71	0.88	0.76	0.55
IRT-1	Sp	0.94	0.97	0.95	0.87	0.94	0.97	0.94	0.98	0.95	0.89	0.93	0.99
	$d_{SeSp}$	0.28	0.39	0.28	**0.18**	0.28	0.39	0.27	0.41	0.29	**0.17**	0.25	0.45
	Se	0.84	0.68	0.75	0.86	0.84	0.66	0.81	0.67	0.77	0.84	0.79	0.58
IRT-2	Sp	0.91	0.98	0.96	0.85	0.94	0.99	0.94	0.99	0.97	0.89	0.95	0.99
	$d_{SeSp}$	**0.18**	0.32	0.25	**0.20**	**0.17**	0.34	**0.20**	0.33	0.23	**0.20**	0.22	0.42

Note: The bold font shows the best results ($d_{SeSp\;}$ ≤ 0.20).

**Table 8 sensors-22-00215-t008:** Maximum permissible errors defined in different standards.

Standards	Devices (Required Minimum Display Range)	Maximum Permissible Errors, in Specific Temperature Ranges	Accuracy Type (Laboratory/Clinical)	Note
IEC 80601-2-59: 2017 [30]	IRTs (None)	±0.5 °C, 34.0–39.0 °C.	Laboratory	Errors from all the test devices are combined.
ISO 80601-2-56: 2017 [34]	clinical thermometers (34.0–43.0 °C)	±0.3 °C, withing the rated output range; ±0.4 °C, withing the rated extended output range.	Laboratory	This standard is under revision for improvement.
ASTM E1112-00: 2011 [58]	electronic thermometers (35.5–41.0 °C)	±0.3 °C, < 35.8 °C; ±0.2 °C, 35.8–37.0 °C; ±0.1 °C, 37.0–39.0 °C; ±0.2 °C, 39.0–41.0 °C; ±0.3 °C, > 41.0 °C.	Not clear	
ASTM E1965-98: 2016 [33]	IR thermometers (Ear canal: 34.4–42.2 °C; Skin: 22.0–40.0 °C)	For ear canal IR thermometers: ±0.3 °C, < 36.0 °C; ±0.2 °C, 36.0–39.0 °C; ±0.3 °C, > 39.0 °C. For skin IR thermometers: ±0.3 °C, over the display range.	Laboratory	

## Data Availability

Not applicable.

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
