# Peer review of "Infrared Thermography for Measuring Elevated Body Temperature: Clinical Accuracy, Calibration, and Evaluation"

_sensors, 2021, doi:10.3390/s22010215_

Round 1

Reviewer 1 Report

please see document attached

Author Response

Dear editor and respected reviewers,

Thank you for the constructive comments. We have carefully and thoroughly addressed each item and modified the paper accordingly. Please see our response below and the revised manuscript for details. All edits in the manuscript are tracked. We believe the paper has been improved because of the changes. We are confident about the quality of our paper and would like to publish all review reports and editorial decisions alongside our paper. So, any comments as well as our response will be considered part of the publication.

Sincerely,

Quanzeng Wang, Ph.D.

Research Biomedical Engineer

Division of Biomedical Physics

Office of Science and Engineering Laboratories

Center for Devices and Radiological Health

U.S. Food and Drug Administration

Reviewer 2 Report

Comments and suggestions are described in the attached file.

Author Response

(The authors gave the same response as above.)

Round 2

Reviewer 1 Report

The authors have provided an adequate and substantiate response to the questions of the reviewer of which several answers could be found in a previous paper of the authors to which they refer many times. The question is if the submitted paper is easily readable without knowledge from the previous paper.   

The authors state that this publication might contribute to the improvement of next-generation standards for IRT systems and the reviewer agrees. As mentioned, the reviewer recommends to emphasize that the new standards should take physiologic mechanisms into account in developing calibration algorithms for IRT systems.